# Targeted cortical reorganization using optogenetics in non-human primates

Azadeh Yazdan-Shahmorad[1,2,3†]*, Daniel B Silversmith[2,4†]*, Viktor Kharazia[2], Philip N Sabes[1,2,4]

[1]Department of Physiology, University of California, San Francisco, San Francisco, United States; [2]Center for Integrative Neuroscience, University of California, San Francisco, San Francisco, United States; [3]Departments of Bioengineering and Electrical Engineering, University of Washington, Seattle, United States; [4]UC Berkeley – UCSF Graduate Program in Bioengineering, University of California, San Francisco, San Francisco, United States

**Abstract** Brain stimulation modulates the excitability of neural circuits and drives neuroplasticity. While the local effects of stimulation have been an active area of investigation, the effects on large-scale networks remain largely unexplored. We studied stimulation-induced changes in network dynamics in two macaques. A large-scale optogenetic interface enabled simultaneous stimulation of excitatory neurons and electrocorticographic recording across primary somatosensory (S1) and motor (M1) cortex (Yazdan-Shahmorad et al., 2016). We tracked two measures of network connectivity, the network response to focal stimulation and the baseline coherence between pairs of electrodes; these were strongly correlated before stimulation. Within minutes, stimulation in S1 or M1 significantly strengthened the gross functional connectivity between these areas. At a finer scale, stimulation led to heterogeneous connectivity changes across the network. These changes reflected the correlations introduced by stimulation-evoked activity, consistent with Hebbian plasticity models. This work extends Hebbian plasticity models to large-scale circuits, with significant implications for stimulation-based neurorehabilitation.

DOI: https://doi.org/10.7554/eLife.31034.001

*For correspondence:
azadehy@uw.edu (AY-S);
dsilversmith@berkeley.edu (DBS)

†These authors contributed equally to this work

## Introduction

Many neurological and psychiatric disorders arise from dysfunctional neural dynamics at the network level, which in turn stem from aberrant neural connectivity (*Stam, 2014*; *Wu et al., 2016*; *Edwardson et al., 2013*; *Yahata et al., 2016*; *DeSalvo et al., 2014*; *Skudlarski et al., 2010*). The brain shows marked plasticity across a variety of learning and memory tasks (*Takeuchi et al., 2014*; *Bliss et al., 2014*) and during recovery after brain injury or stroke (*Edwardson et al., 2013*; *Hara, 2015*; *Murphy and Corbett, 2009*), and many have proposed to take advantage of this innate plasticity to treat neural disorders (*Miller et al., 2010*; *Jackson et al., 2006*; *Lucas and Fetz, 2013*; *Edwardson et al., 2013*). In principle, brain stimulation protocols can be designed to leverage this plasticity in order to rewire aberrant neural connectivity, potentially curing these disorders. Implementing such treatments requires a better understanding of how stimulation-induced plasticity drives changes in network connectivity and network dynamics.

The simple Hebbian model of plasticity (*Donald, 1949*) and spike-timing dependent versions of it (*Bi and Poo , 2001*) explain a large body of data on activity-dependent plasticity, including both in vitro (*Massobrio et al., 2015*; *Abrahamsson et al., 2016*), and in vivo (*Andersen et al., 2017*; *Feldman, 2012*; *Shulz and Jacob, 2010*) studies. Despite the extensive work studying Hebbian plasticity at the cellular level, it remains unclear how synaptic plasticity leads to large-scale functional reorganization. Recently, several studies have shown large-scale plasticity following brain stimulation that is

**eLife digest** From riding a bike to reaching for a cup of coffee, all skilled actions rely on precise connections between the sensory and motor areas of the brain. While sensory areas receive and analyse input from the senses, motor areas plan and trigger muscle contractions. Precisely adjusting the connections between these and other areas enables us to learn new skills, and it also helps us to relearn skills lost as a result of brain injury or stroke.

About 70 years ago, a psychologist named Donald Hebb came up with an idea for how this process might occur. He proposed that whenever two neurons are active at the same time, the connection between them becomes stronger. This idea, that 'cells that fire together, wire together', became known as Hebb's rule. Many studies have since shown that Hebb's rule can explain changes in the strength of connections between pairs of neurons. But can it also explain how connections between entire brain regions become stronger or weaker?

New results show that it can. The data were obtained using a technique called optogenetics, in which viruses are used to introduce genes for light-sensitive proteins into neurons. Shining light onto the brain will then activate any cells within that area that contain the resulting proteins. Yazdan-Shahmorad, Silversmith et al. used this technique to activate small regions of either sensory or motor brain tissue in live macaque monkeys. Doing so strengthened the overall connectivity between the two areas. The effects were more variable at the level of smaller brain regions, with some connections becoming weaker rather than stronger. However, Yazdan-Shahmorad, Silversmith et al. show that Hebb's rule explains most of the observed changes.

Many neurological and psychiatric disorders stem from abnormal brain connectivity. Simple forms of brain stimulation are already used to treat certain neurological disorders, such as Parkinson's disease. Stimulating the brain to induce specific changes in connectivity may ultimately enable us to leverage the brain's natural learning mechanisms to cure, instead of just treat, these conditions.
DOI: https://doi.org/10.7554/eLife.31034.002

consistent with Hebbian mechanisms (*Jackson et al., 2006*; *Lucas and Fetz, 2013*; *Seeman et al., 2017*; *Nishimura et al., 2013*; *Rebesco and Miller, 2011*; *Rebesco et al., 2010*; *Song et al., 2013*; *Lajoie et al., 2017*). In particular, Fetz and colleagues implemented an activity-dependent stimulation protocol, effectively introducing an artificial connection between two sites in the motor cortex (*Jackson et al., 2006*; *Lucas and Fetz, 2013*). Continuous reinforcement of this artificial connection led to a stable functional change in stimulation-evoked movements, indicating that stimulation induces large-scale plasticity. Recently these results were reproduced in a modeling work at the network level (*Lajoie et al., 2017*). Similar results have been observed using open-loop stimulation protocols to induce targeted plasticity between two cortical sites (*Rebesco and Miller, 2011*; *Seeman et al., 2017*). Notably, these papers have reported off-target effects, but these were either interpreted as global changes in excitability or were not explained. Although the results from closed-loop and open-loop experiments suggest large-scale plasticity, the underlying neural network changes remain unexplored.

Here, we measure connectivity across sensorimotor cortex and track changes in network connectivity in response to open-loop stimulation. This work takes advantage of a large-scale optogenetic interface (*Yazdan-Shahmorad et al., 2016*) that enables us to simultaneously stimulate populations of excitatory neurons while recording large-scale μ-electrocorticography (μ-ECoG) activity across two brain areas, S1 and M1. We first establish and compare two measures of functional connectivity that provide complementary views of the mechanisms of plasticity. Next, we investigate how stimulation impacts connectivity between and within cortical areas. We then test whether stimulation-evoked activity drives large-scale network plasticity in a Hebbian manner. The goal of this work is to investigate large-scale functional reorganization following stimulation, which will inform future neuro-rehabilitation strategies.

## Results

We performed optogenetic stimulation via laser illumination of the cortical surface while simultaneously recording surface potentials (μECoG) from about 1.5 cm$^2$ of primary somatosensory (S1) and

**Figure 1.** Stimulation and recording setup. Photo of the µECoG array placement over M1 and S1 in Monkey G (left panel) and the placement of lasers on top of the array in two different configurations (middle panel: Monkey G and right panel: Monkey J).

DOI: https://doi.org/10.7554/eLife.31034.003

The following figure supplement is available for figure 1:

**Figure supplement 1.** Opsin expression was observed only in pyramidal neurons.
DOI: https://doi.org/10.7554/eLife.31034.004

motor (M1) cortices (*Figure 1*). The viral vector used to obtain opsin expression targeted excitatory neurons (see supplementary material for details; *Figure 1—figure supplement 1*). These neurons have strong projections within and between the two brain areas (*Murray and Keller, 2011*; *Kinnischtzke et al., 2014*; *Weiler et al., 2008*). Activating these excitatory cells should increase the excitability of the underlying network, improving the likelihood of neuroplastic change (*Iriki et al., 1989*). In two macaque monkeys we explored stimulation-induced changes in network connectivity.

## Two measures of functional connectivity between M1 and S1 show strong correlation

We quantified inter-area functional connectivity between S1 and M1 using two different measures— one based on the network response to optogenetic stimulation and the other based on spontaneous neural activity.

The first measure focused on how the response to optical stimulation propagates through the network. Optogenetic stimulation in S1 and M1 evoked responses across both cortical areas, and we characterized these responses according to their amplitudes and delays (*Figure 2A and E*). The delays exhibited a bimodal distribution with a 3–6 ms separation between the early and late responses (*Figure 2B and F*). Classifying the responses across electrodes by delay recovers the spatial separation between the two cortical areas—following a boundary along the central sulcus—with shorter delays in the area being stimulated (S1 or M1; primary responses) and longer delays in the other area (M1 or S1; secondary responses) (*Figure 2C and G*). The short delay of the secondary responses suggests close functional connectivity between these areas. We quantified this connectivity with the stimulus-evoked response ratio (SERR). SERR was defined as the peak-to-trough amplitude of the secondary responses (A2) normalized by the peak-to-trough amplitude of the primary response (A1), with both responses measured in the high gamma (60–200 Hz) filtered signal (see dashed-line inset boxes in *Figure 2*,A and E). The SERR is a measure of the connectivity between the site of stimulation and sites in the other cortical area (see *Figure 2D and H*).

The second measure evaluates functional connectivity during spontaneous activity. We focused on the coherence in field potential recordings between electrodes, a widely used measure of connectivity that captures the degree of phase-locking between two signals (*Lang et al., 2012*; *Bastos and Schoffelen, 2015*). We calculated the coherence between the site of stimulation and the secondary sites across different frequency bands.

Because SERR and coherence are derived independently from recordings with and without simultaneous stimulation, they might reflect fundamentally different aspects of network connectivity. Therefore, in each experiment we compared these two measures prior to conditioning stimulation. *Figure 3A* shows two examples comparing SERR to coherence in the theta band (4–8 Hz). To quantify the relationship between these measures, we performed linear regression between them across channels (*Figure 3B–C*). An example showing a strong relationship between SERR and theta band

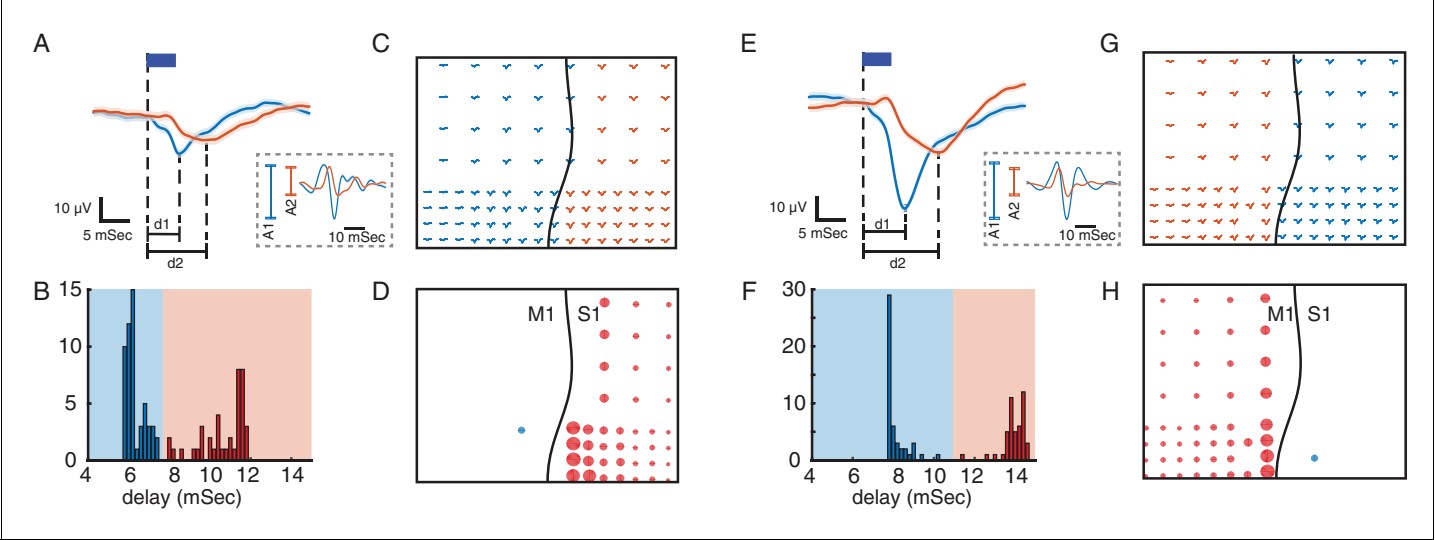

**Figure 2.** Using evoked responses to measure connectivity across M1 and S1. (**A**) Primary (blue) and secondary (orange) evoked responses to light stimulation. The dark blue rectangle represents the duration of light stimulation. Shaded areas show standard error. The delays of evoked responses are calculated as the time difference between the onset of stimulation and the time of the response trough (vertical lines at d1 and d2). (**B**) Distribution of the delays color-coded based on the primary and secondary light-evoked responses. (**C**) Evoked responses across the array, color-coded based on the delays. As shown here there is a spatial separation of primary and secondary responses that corresponds to the locations of M1 and S1. This suggests that due to functional connectivity between M1 and S1, we see a delayed (secondary) response in S1 to light stimulation in M1. (**D**) S1 connectivity with the site of stimulation. The blue circle shows the location of stimulation in M1. The black line shows the location of central sulcus with respect to the recording array. The size of the red circles represents the strength of connectivity between each site and the stimulation location for the recording sites with secondary responses across S1. SERR Connectivity is defined as the peak-to-trough of filtered high gamma responses (60–200 Hz: trace plots shown on the dashed rectangle on A) at each site (orange) normalized to the peak-to-trough of high gamma at the site of stimulation (blue). (**E–H**) Same as A-D with the same array placement and S1 stimulation.

DOI: https://doi.org/10.7554/eLife.31034.005

coherence is shown in *Figure 3B*. This analysis was repeated for all coherence frequency bands and for all experiments; the mean and standard error of the regression parameters are shown in *Figure 3C*. The two measures of functional connectivity were highly correlated: across sessions, the distribution of regression slopes was significantly different from 0 (t-test: p=3.73e-06), and 69 percent of individual experiments showed significant linear regressions (p<0.05). This finding indicates that at baseline, before any conditioning, the stimulation-evoked response reflects network dynamics across the frequency spectrum.

## Stimulation strengthens inter-area connectivity

We used a simple stimulation protocol—we delivered 5 ms laser light pulses at a frequency of 5 or 7 Hz at either one or two cortical sites (*Figure 4A*). For our initial analyses, and unless otherwise noted, when stimulating at two cortical sites, we alternated stimulation between the two light sources to avoid any interference between the evoked responses from each light source. In each experiment, conditioning stimulation was applied for 50 min, and we evaluated functional connectivity every 10 min during blocks of passive baseline recording and active testing (100 light pulses delivered through each laser). We also conducted control sessions with the same passive recording and testing blocks, but with no stimulation during the conditioning blocks.

A representative example experiment, shown in *Figure 4B*, shows that stimulation in M1 leads to significant increases (paired t-test, p=3.88e$^{-11}$) in mean SERR. Similar increases in SERR were observed for a majority of experiments (14 out of 18 experiments) across both monkeys (*Figure 4C*). The increases in connectivity were symmetric between the two cortical areas (*Figure 4D*). No significant change in mean SERR was observed in control sessions without conditioning stimulation (*Figure 4D*). Furthermore, the increase in SERR was significantly larger for stimulation sessions than control sessions (unpaired t-test: p=0.0036)

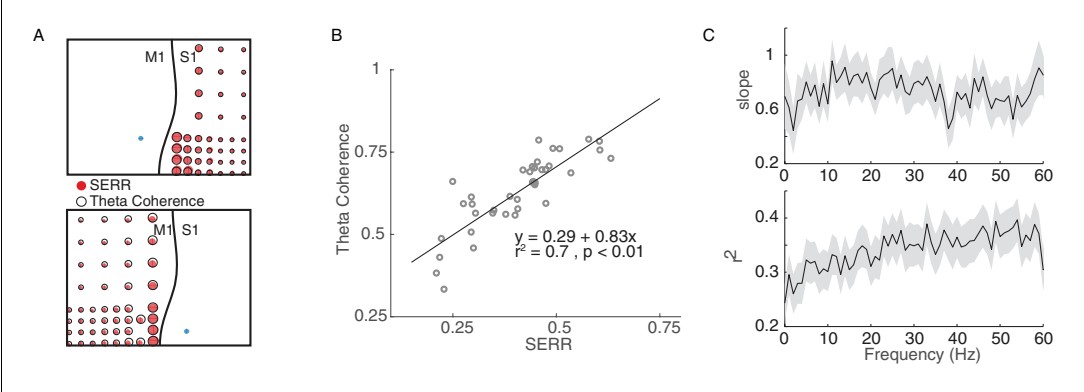

**Figure 3.** Coherence measure of inter-area connectivity correlates with SERR. (**A**) S1 (Top panel) and M1 (bottom panel) theta coherence and SERR connectivity with the site of stimulation in monkey G. The blue circle shows the location of stimulation. The black line shows the location of central sulcus with respect to the recording array. The size of the red and white circles represents the strength of connectivity between each secondary site and the stimulation location. (**B**) An example session showing relationship between SERR and theta coherence. The black line shows the linear regression fit. (**C**) Linear relationship between evoked response and coherence across different frequencies. Summary data showing the mean and standard error (shaded region) of regression parameters (shown in B) across frequencies.

DOI: https://doi.org/10.7554/eLife.31034.006

The following source data and source codes are available for figure 3:

**Source code 1.** Comparing SERR and coherence measurements for example session and across sessions.
DOI: https://doi.org/10.7554/eLife.31034.007
**Source data 1.** SERR and coherence across channels for example session and across sessions.
DOI: https://doi.org/10.7554/eLife.31034.008

We next asked whether conditioning increased coherence-based measures of inter-area connectivity. We did observe increases in mean theta band (4–8 Hz) coherence following conditioning, as shown in *Figure 4E* for the same dataset used in *Figure 4B* (paired t-test, p=0.03). The increase in theta coherence was significant for the majority of experiments (13 out of 20; *Figure 4E*), and the effect was localized to the theta band (*Figure 4G*; paired t-test, p=0.017; Bonferroni correction for multiple comparisons). We also looked for change in the theta-band power as a result of stimulation and did not see any significant changes, supporting the conclusion that changes in coherence reflect changes in functional connectivity (See *Figure 4—figure supplement 2*).

## Stimulation weakens correlation between measures of functional connectivity

We showed in *Figure 3C* above that SERR and coherence are correlated measures of functional connectivity in the baseline condition. Here we ask how this relationship is changed by stimulation. *Figure 4H* (black lines) replicates the baseline data from *Figure 3C*, along with new data showing the correlation between these measures at the end of each experiment (green lines) and between their conditioning-induced changes (yellow lines). After conditioning, the cross-channel correlation between these measures significantly decreased (paired t-test, p=6.3e-10), and the changes in connectivity were uncorrelated (t-test on distribution of regression slopes, p=0.85). This suggests that the stimulation affects the two measures differently.

To further address this, we replicated this analysis for our control (no stimulation) sessions. In this case, we did not see a significant decrease in the correlation between measures after control sessions (paired t-test, p=0.36; see *Figure 4—figure supplement 1*). Additionally, a direct comparison between the median decrease in the correlation for stimulation versus control sessions showed a significant difference when all frequencies were considered (p=0.0025), though the difference was not significant when only compared in the theta band. Together, these results show that the stimulation-induced changes in these two measures of connectivity do not correlate with each other at a fine-scale. This may reflect the fact that the two measures are not functionally equivalent, despite their correlation, i.e., they capture related, but not identical, aspects of connectivity.

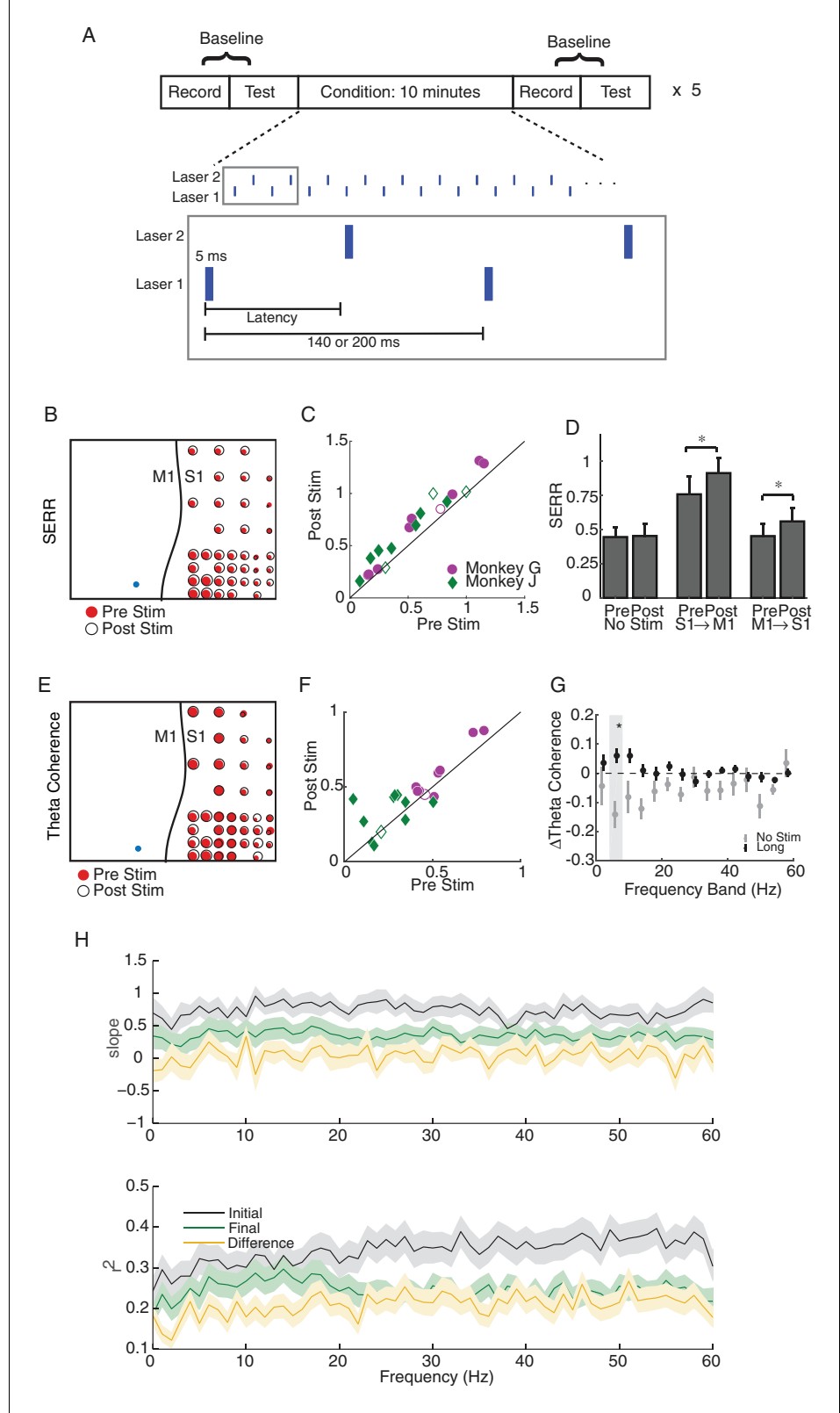

**Figure 4.** Single site and long-latency stimulation increase the functional connectivity between M1 and S1. (**A**) Experimental protocol. Conditioning stimulation was interrupted by periodic connectivity measurements including passive recording and active testing. Either one or two non-interfering lasers were used. (**B**) Examples of changes in SERR across the recording array in monkey G. Red circles show initial connectivity and white circles show

*Figure 4 continued on next page*

*Figure 4 continued*

connectivity after 50 min. of conditioning. (**C**) Summary of SERR changes across all experiments. Each symbol represents the connectivity averaged over all secondary channels for one experiment. Filled markers show significant changes (paired t-test; p<0.05, the p-values for each experiment are listed in the supplementary spreadsheet). (**D**) Changes in SERR when stimulating in either S1 or M1 in comparison to control. Error bars represent standard error and asterisks show significant changes (paired t-test; Control: p=0.8, laser in S1: p=0.01, laser in M1: p=1.6e-04). (**E**) Examples of changes in theta (4–8 Hz) coherence across the recording array. (**F**) Summary of theta coherence changes across all experiments. Each marker represents the connectivity averaged over all channels for one experiment. Filled markers show significant changes (paired t-test; p<0.05, the p-values for each experiment are listed in the supplementary spreadsheet). (**G**) Change in coherence across different frequency bands in comparison to controls. Asterisk show significant difference between the two groups (unpaired t-test, Bonferroni corrected; p<0.05, the p-values are listed in the supplementary spreadsheet). (**H**) Linear relationship between SERR and coherence across different frequencies for pre-stim, post-stim and the change in both measures. Summary data showing the mean and standard error (shaded region) of regression slope and r² as a function of coherence frequency (see the example regression for the theta-band in *Figure 3B*).
DOI: https://doi.org/10.7554/eLife.31034.009

The following source data, source code and figure supplements are available for figure 4:

**Source code 1.** Comparing SERR and coherence measurements across channels for example session and for all sessions broken down by experimental condition.
DOI: https://doi.org/10.7554/eLife.31034.016

**Source data 1.** SERR and coherence measurements across channels for example session and for all sessions broken down by experimental condition.
DOI: https://doi.org/10.7554/eLife.31034.017

**Figure supplement 1.** Linear relationship between SERR and coherence across different frequencies for control sessions.
DOI: https://doi.org/10.7554/eLife.31034.010

**Figure supplement 1—source code 1.** SERR and coherence relationship across different frequencies for control sessions.
DOI: https://doi.org/10.7554/eLife.31034.011

**Figure supplement 1—source data 1.** SERR and coherence measurements acrossdifferent frequencies and channels for control sessions.
DOI: https://doi.org/10.7554/eLife.31034.012

**Figure supplement 2.** Change in power following stimulation.
DOI: https://doi.org/10.7554/eLife.31034.013

**Figure supplement 2—source code 1.** Comparison of power measurements between control and stimulation data across different frequency bands.
DOI: https://doi.org/10.7554/eLife.31034.014

**Figure supplement 2—source data 1.** Power measurementsacross different frequency bands for each session.
DOI: https://doi.org/10.7554/eLife.31034.015

## Stimulation drives increases in inter-area connectivity within 10 min blocks

As shown in *Figure 1B*, our stimulation protocol included 5 repetitions of baseline recording, testing, and conditioning. This design allowed us to track connectivity changes across time, after each 10 min increment of conditioning. Additionally, since neural activity was recorded throughout the experiment, we were able to measure changes in SERR during conditioning blocks. *Figure 5A* shows the evolution of mean inter-area connectivity for both measures in an example session. To account for differences in network connectivity across monkeys and sessions, we also analyzed the changes in each measure with respect to the baseline pre-conditioning measurements (*Figure 5B*). For this example session (*Figure 5A*), and on average across all sessions (*Figure 5B*), there is a trend of increasing connectivity across the session that starts within the first 10 min conditioning block. These increases in inter-area connectivity measured with SERR and coherence (*Figure 5B*) were correlated across time (Pearson correlation coefficient = 0.26). Additionally, the rate of strengthening was consistent across conditioning blocks; we detected a significant increase in SERR (p<0.05) at ~ 4.5 min of stimulation in 4 out of 5 conditioning blocks (see *Figure 5—figure supplement 1* for comparison with control sessions).

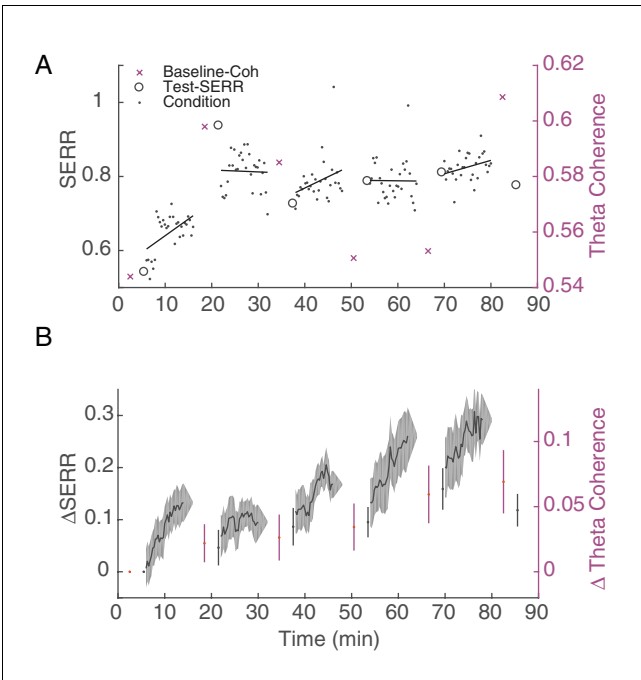

**Figure 5.** Stimulation induces an increase in inter-area connectivity across time. (**A**) An example of the dynamics of mean inter-area SERR and coherence in an experiment. Black lines show linear regression to each conditioning block. (**B**) Dynamics of change in SERR and theta coherence with respect to baseline connectivity across all experiments (shaded area show standard error).

DOI: https://doi.org/10.7554/eLife.31034.018

The following source data, source code and figure supplements are available for figure 5:

**Source code 1.** Change in SERR and coherence measurements for example session and summary over stimulation sessions.
DOI: https://doi.org/10.7554/eLife.31034.022
**Source data 1.** SERR and coherence measurements for stimulation sessions.
DOI: https://doi.org/10.7554/eLife.31034.023
**Figure supplement 1.** Dynamics of changes in connectivity for control sessions.
DOI: https://doi.org/10.7554/eLife.31034.019
**Figure supplement 1—source code 1.** Change in SERR and coherence measurements for control sessions across time.
DOI: https://doi.org/10.7554/eLife.31034.020
**Figure supplement 1—source data 1.** SERR and coherence measurements for control sessions across time.
DOI: https://doi.org/10.7554/eLife.31034.021

Note that the stimulation-induced plasticity was not reinforced during baseline recording blocks. Given the length of the conditioning (10 min) and baseline recording (5 min in most experiments) blocks, one might expect that stimulation-induced changes would revert back to baseline. We did observe a significant decrease (paired t-test, p=2.9e-05) in the SERR between the last 100 pulses of the conditioning block and the following test block, indicating some unlearning of the stimulation-induced connectivity changes.

## Stimulation drives fine-scale changes across the network that are consistent with Hebbian learning

We next evaluated the fine-scale effects of conditioning across the entire network. Since SERR is restricted to connectivity from the stimulation site, this analysis was conducted using only changes in pairwise coherence. *Figure 6A* shows data from an example session, with connectivity changes represented as a set of heatmaps: the heatmap at each recording site represents the change in pairwise

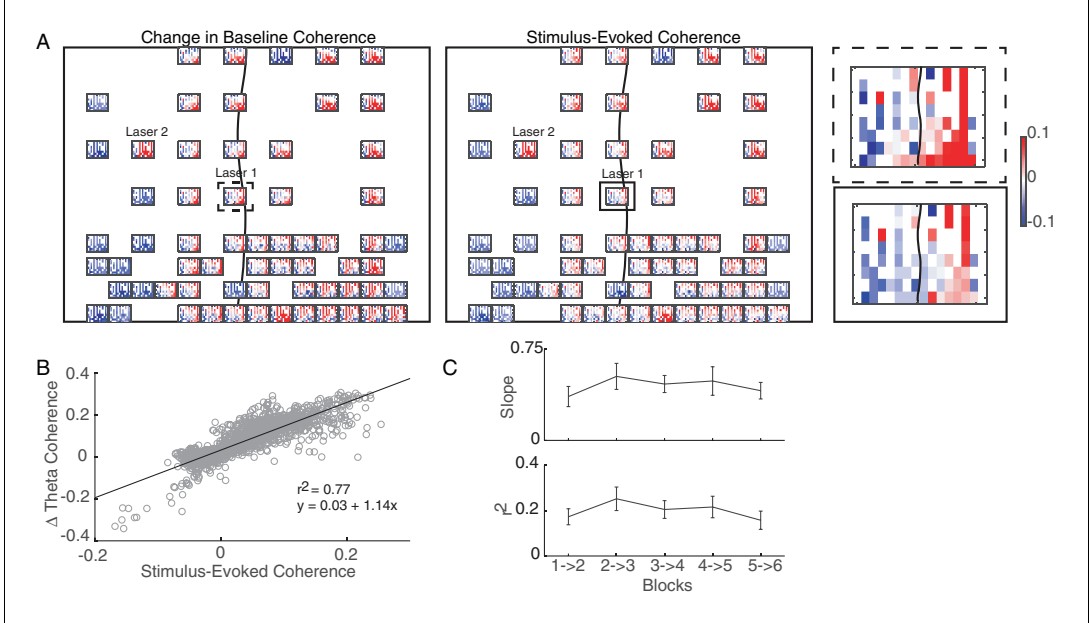

**Figure 6.** Hebbian plasticity models explain stimulation-induced fine-scale network connectivity changes. (**A**) An example session highlighting the similarity between the stimulus-evoked coherence (middle panel) and the change in baseline coherence (left panel) across the array in Monkey J. At each recording site across the array, a heatmap represents the respective coherence between that location with all other recording sites. Enlarged examples for a single location are compared in the right panel. The black line shows the location of central sulcus on the array and rectangles show the location of the magnified examples. The average value across the array was subtracted for visualization. (**B**) Linear regression between stimulus-evoked coherence and the change in baseline coherence for the example session shown in A. (**C**) Summary of regression parameters across single-site and non-interference experiments. Errorbars show standard error.

DOI: https://doi.org/10.7554/eLife.31034.024

The following source data and source codes are available for figure 6:

**Source code 1.** Comparing pairwise coherence measurements for example session and summary of regression parameters for stimulation sessions.

DOI: https://doi.org/10.7554/eLife.31034.025

**Source data 1.** Pairwise coherence measurements for example session and regression parameters for all sessions by experimental condition.

DOI: https://doi.org/10.7554/eLife.31034.026

coherence between that electrode and all of the other electrodes. A magnified example heatmap is shown in *Figure 6A*, right; top panel. This example shows heterogeneous changes in connectivity across the network.

We next asked whether such heterogeneous changes are to be expected. A Hebbian plasticity model suggests that the fine-scale changes in connectivity should reflect the statistics of plasticity-inducing activity. In this case, we predicted that correlated stimulation-evoked activity between two cortical sites would strengthen the functional connectivity between them, while uncorrelated or anti-correlated stimulation-evoked activity would weaken their functional connectivity. We quantified stimulation-induced correlations by measuring the pairwise coherence during the conditioning block and subtracting out the coherence calculated during baseline recording (thought to be representative of initial connectivity). We refer to this difference as the stimulus-evoked coherence. Indeed, in the example session of *Figure 6A*, one can see a similarity between the pattern of changes in baseline coherence across the array (left panel) with the stimulus-evoked coherence (middle panel).

To quantify this similarity, we used linear regression to predict changes in baseline coherence across sites based on the stimulus-evoked coherence. Importantly, both of these quantities use the initial baseline coherence as a reference. So to avoid spurious correlations introduced by subtracting the same data from the two regression variables, we split the baseline recording blocks into two intervals and used the data from only one interval for each regressor (See Materials and Methods for more details). Data from the example session in *Figure 6A* is shown as a scatter plot in *Figure 6B*, along with the calculated regression parameters. The plot shows a strong correlation between the stimulus-evoked coherence and changes in baseline coherence. This relationship was consistent

across our data (*Figure 6C*): the distribution of linear regression slopes across experimental sessions and monkeys was significant (paired t-test: p=2.41e-19; 101 out of 105 experimental blocks had significant (p<0.05) linear regression models). These results support a Hebbian model for network-wide changes in connectivity.

To further test this model, we conducted some experimental sessions using a more complex spatio-temporal pattern of optical stimulation. Specifically, by reducing the latency between the two light sources to either 10 or 30 ms (*Figure 7A*), we introduce interference between evoked responses from the two light sources (*Figure 7B*). We repeated the analysis of mean inter-area coherence changes (c.f. *Figure 4F*) for these sessions, and we did not see a consistent increase in mean inter-area connectivity (*Figure 7C*). Nevertheless, we found that stimulus-evoked coherence continued to predict changes in baseline coherence (paired t-test on the distribution of regression slopes: p=1.37e-28; linear regression slopes with p<0.05: 160 out 170 blocks; *Figure 7D*, see *Figure 7—figure supplement 1* for an example session). Furthermore, the regression parameters were similar to those obtained with the simple stimulation patterns (*Figure 6C*), and both simple and complex stimulation experiments yielded regression slopes that were significantly larger than those obtained from control sessions (long-latency vs. control sessions, ranksum test: p=0.048; short-latency vs. control sessions, ranksum test: p=0.037; see *Figure 7—figure supplement 2*).

These results demonstrate a robust relationship between stimulus-evoked correlations and changes in baseline connectivity across the network. These findings are well explained by a Hebbian model of large-scale stimulation-induced plasticity.

## Discussion

In this study we investigated how the functional connectivity between and across sensory and motor areas changes in response to stimulation. With optogenetics, we selectively manipulated local populations of excitatory neurons within this sensorimotor network. We compared two different methods to measure functional connectivity at both gross- and fine-scales, and demonstrated how these measures change in response to conditioning stimulation. We showed that these changes are consistent with Hebbian synaptic plasticity rules, extending Hebbian models of stimulation-driven plasticity to large-scale networks. This work demonstrates the feasibility of driving targeted plasticity with optogenetic stimulation. This framework is a starting point for designing principled approaches to large-scale neuroplasticity and stimulation-based therapies for neurological and neuropsychiatric disorders.

### Measures of Functional connectivity between M1 and S1

Functional connectivity between M1 and S1 has been demonstrated in fMRI (*Matsui et al., 2011*; *McGregor and Gribble, 2015*), electrophysiology (*Iriki et al., 1989*; *Yazdan-Shahmorad et al., 2016*; *Prsa et al., 2017*), and anatomical studies (*Kosar et al., 1985*; *Petrof et al., 2015*). To evaluate stimulation-induced changes across these areas we investigated two measures of functional connectivity, one based on stimulation and one based on natural neural processing. Variants of SERR (*Seeman et al., 2017*; *Feldmeyer and Sakmann, 2000*; *Klinshov et al., 2014*) and coherence (*Bastos and Schoffelen, 2015*; *Lang et al., 2012*) have been used previously to evaluate functional connectivity. However, to our knowledge they have never been used in combination or compared.

In principle, these two measures reflect different aspects of functional connectivity. SERR is a more direct measure of the projections from the stimulation site. Importantly, stimulation likely evokes activity from both terminals and cell bodies located at the stimulation site, though C1V1 does not express well down at the axon (*Rajasethupathy et al., 2015*). Secondary responses arise from a combination of synaptic, antidromic, and indirect (network) effects. That said, based on the timing of the evoked responses and the limited expression of C1V1 in axon terminals, we expect the amplitude of secondary responses normalized to primary response amplitudes to reflect synaptic connectivity. Coherence, on the other hand, measures the broader effects of network-wide dynamics.

Despite these differences, we saw that the measures are robustly correlated across channels at the beginning of our experiments. This would be expected in a network where stability was maintained with a Hebbian mechanism. In other words, direct connectivity, as reflected in SERR, drives correlated activity, which is reflected in coherence; conversely, correlated activity drives changes in

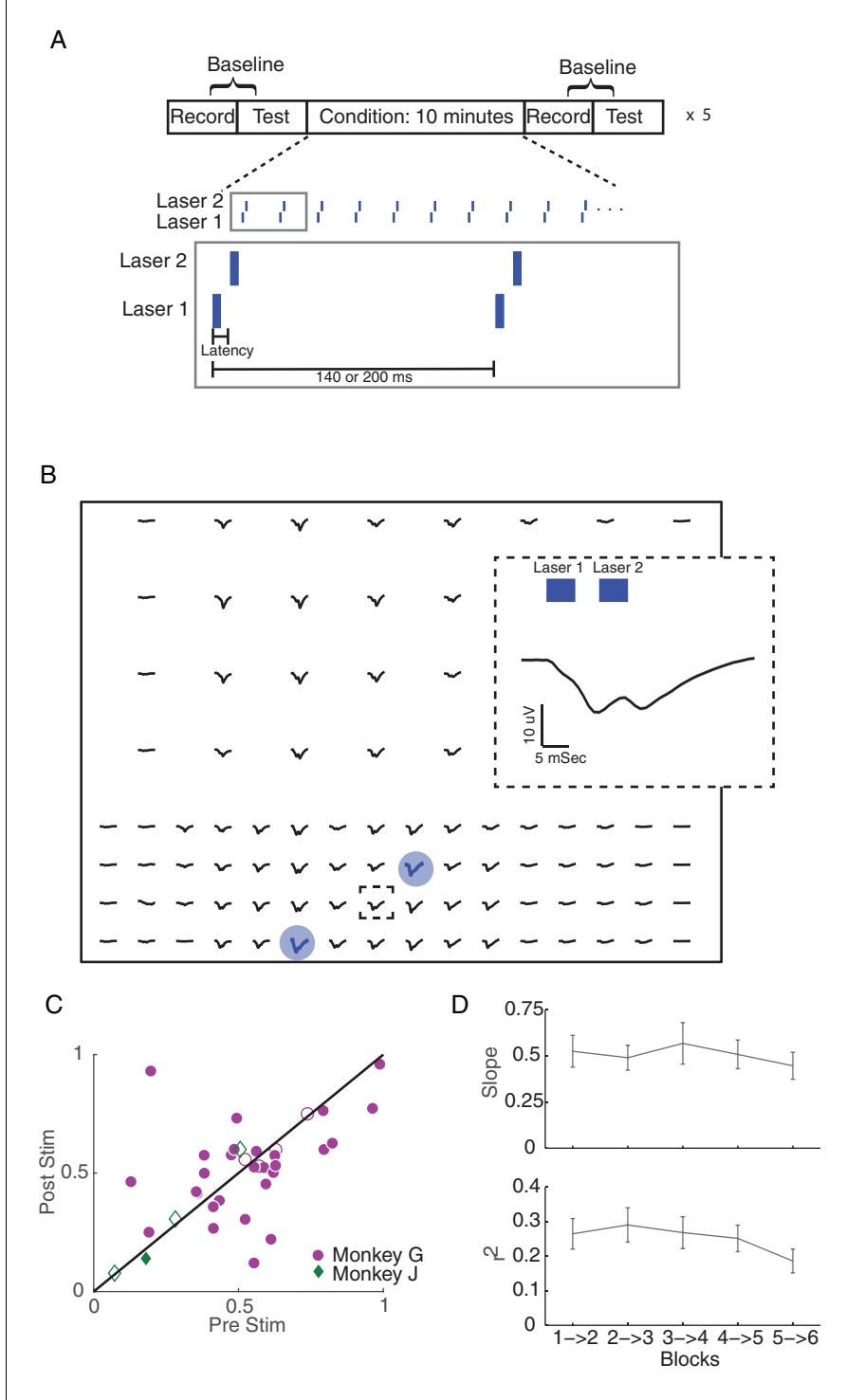

**Figure 7.** Hebbian plasticity models explain fine-scale network connectivity changes driven by complex spatio-temporal stimulation patterns. (**A**) Same stimulation protocol as described in 4A. However, here we reduced the latency between to the two lasers to 10 ms or 30 ms, creating a more complicated pattern of stimulation. (**B**) An example of the stimulus-evoked activity across the array from Monkey G. Blue circles show the locations of stimulation. The inset shows the enlarged pattern of evoked response at the framed electrode, which is located close to both lasers. (**C**) Summary of inter-area theta coherence changes for all interference experiments across both monkeys. (**D**) Summary of regression parameters across all interference experiments. Errorbars show standard error.

*Figure 7 continued on next page*

*Figure 7 continued*

DOI: https://doi.org/10.7554/eLife.31034.027

The following source data, source code and figure supplements are available for figure 7:

**Source code 1.** Coherence measurements and regression parameters for short-latency stimulation sessions.
DOI: https://doi.org/10.7554/eLife.31034.034

**Source data 1.** Coherence measurements and regression parameters for short-latency stimulation sessions.
DOI: https://doi.org/10.7554/eLife.31034.035

**Figure supplement 1.** Short-latency example showing the relationship between input coherence and change in baseline coherence.
DOI: https://doi.org/10.7554/eLife.31034.028

**Figure supplement 1—source code 1.** Comparing pairwise coherence measurements for example session and regression parameters for all sessions by experimental condition.
DOI: https://doi.org/10.7554/eLife.31034.029

**Figure supplement 1—source data 1.** Pairwise coherence measurements for example session and regression parameters for all sessions by experimental condition.
DOI: https://doi.org/10.7554/eLife.31034.030

**Figure supplement 2.** Comparing the effects of stimulation on network connectivity between stimulation and controls.
DOI: https://doi.org/10.7554/eLife.31034.031

**Figure supplement 2—source code 1.** Plot regression parameters for all sessions by experimental condition.
DOI: https://doi.org/10.7554/eLife.31034.032

**Figure supplement 2—source data 1.** Regression parameters for all sessions by experimental condition.
DOI: https://doi.org/10.7554/eLife.31034.033

connectivity (see *Network effects*). These coupled mechanisms lead the network to a stable, steady state (*Fox and Stryker, 2017*; *Toyoizumi et al., 2014*; *Zenke and Gerstner, 2017*).

Notably, however, our conditioning protocol caused the two measures of connectivity to become less correlated across electrodes. This might seem surprising given that conditioning led to a significant mean increase in connectivity between S1 and M1 for both measures and that the changes in coherence were robustly predicted by the stimulus-evoked coherence. However, this decrease in the correlation between the measures after conditioning seems to be limited to high frequencies and may reflect a transient instability of the network due to the stimulation-induced perturbations in connectivity.

## Stimulation changes the functional connectivity between M1 and S1

Previous work has demonstrated that either activity dependent (*Jackson et al., 2006*; *Lucas and Fetz, 2013*; *Nishimura et al., 2013*; *Rebesco et al., 2010*; *Song et al., 2013*) or paired electrical stimulation (*Rebesco and Miller, 2011*; *Seeman et al., 2017*) can lead to plastic changes in both primate and rodent cortex. Here we demonstrated that optogenetic stimulation at even a single location strengthens inter-area functional connectivity between brain areas. A similar observation was reported for one session of single-site electrical stimulation (*Seeman et al., 2017*). This result is consistent with spike-timing dependent plasticity (STDP) rules (*Bi and Poo , 2001*). In particular, since the time difference between the activation of the primary (pre-synaptic) and secondary (post-synaptic) responses to optical stimulation is about 3–6 ms, repetitive stimulation should strengthen the connectivity from the site of stimulation to the other area. While stimulation-induced long-term potentiation (LTP) and STDP have been observed within local circuits (*Iriki et al., 1989*; *Feldman, 2012*; *Shulz and Jacob, 2010*), and have been posited as the mechanism of stimulation-induced plasticity within M1 (*Lajoie et al., 2017*) this is the first study explaining large-scale changes across cortical networks. Notably, conditioning drove increases in coherence between S1 and M1 only in the theta band (4–8 Hz). This result is difficult to interpret because the conditioning stimulation frequency (5 or 7 Hz) itself lies within the range of the theta band. It is possible that Hebbian mechanisms are selectively enhancing connectivity at the conditioning frequency. It's also possible that theta band coherence best reflects the plastic connections between these areas. Since we did not vary stimulation frequency outside of this range, no strong conclusions can be drawn.

## Dynamics of connectivity changes

The temporal dynamics of both measures reveal a trend of increasing inter-area connectivity throughout the experiment. This trend suggests that neuroplastic changes start almost immediately after the start of stimulation. Interestingly, we observed unlearning of the stimulation-induced changes during passive recording, though some of the stimulation-induced changes persisted. Previous studies have shown that longer conditioning sessions (3–48 hr) result in changes that are stable over timescales comparable to the length of conditioning (*Jackson et al., 2006*; *Lucas and Fetz, 2013*; *Seeman et al., 2017*; *Nishimura et al., 2013*; *Rebesco and Miller, 2011*). Further experiments are required to estimate the stability of plasticity following optogenetic stimulation over longer time scales.

## Network effects

Given that Hebbian rules have been used extensively to explain synaptic plasticity, we wanted to explore the idea that large-scale stimulation-induced changes are consistent with Hebbian plasticity. The strongest test of the Hebbian model is to compare the correlations directly induced by stimulation with the changes in network correlations observed after conditioning. Our ability to record artifact-free signals during stimulation allowed us to perform this test. We did find a predictive linear relationship between stimulus-evoked coherence and the changes in baseline coherence after conditioning. Furthermore, the model held for both simple and complex spatiotemporal patterns of stimulation-evoked activity.

## Comparison with previous sensory cortical plasticity literature

There is a large body of research on sensory cortical plasticity (*Merzenich et al., 1983*; *Qi et al., 2011*). This study differs in two key dimensions. First, previous work has largely focused on how changes in the statistics of sensory input and/or motor output affect the cortical representation in sensory cortex. In contrast, we have directly manipulated cortical circuits, without a detailed knowledge of the somatotopic map. Also, earlier work studied plasticity on the time scale of days to months, while we are studying changes across minutes. Despite these differences, both appear to support a key role for Hebbian learning in sensorimotor cortical plasticity. Still, a deeper understanding of the relationship between these bodies of work would require a study of intermediate timeframes as well, along with detailed cortical mapping. These are important questions for future work.

## Limitations of the setup

The large-scale interface used in this study enabled robust estimation of functional connectivity and stimulation-evoked correlations. However, there are several limitations in our experimental setup. First, surface stimulation combined with high blue light absorption and scattering in tissue limits the depth of light penetration (*Yizhar et al., 2011*). The ability to target specific cortical layers would help us understand the anatomical basis of the plasticity we observed. Second, surface µECoG recordings reflect a summation of nearby neural activity, making comparisons to explicit synaptic learning rules difficult. Depth recordings would reveal more detailed information about spike timing, the role of different cortical layers, and the relationship between synaptic and large-scale plasticity. Lastly, the µECoG arrays were placed acutely for each experiment. To investigate the long-term effects of stimulation we need stable, chronic recordings.

## Clinical applications

Our results offer a proof-of-concept that optogenetic stimulation can drive predictable changes in network-scale connectivity. The success of optogenetics in NHP represents an important step forward for translational use (*Doroudchi et al., 2011*; *Han, 2012*). The continued rapid progress in the field has the potential to bring cell-type specific neuromodulation therapies to the clinic. However, even where therapeutic approaches ultimately wind up using electrical stimulation, the advantages of optogenetics make it a powerful tool for better understanding the underlying mechanisms of plasticity and, thus, for the development of therapeutic applications of neuromodulation. This framework also provides scientific insight into the mechanisms of neural plasticity. Future efforts should focus on linking the plasticity results presented here with improvements in motor function or sensory perception, which will have important implications for stimulation-based therapies.

## Materials and Methods

Two adult male rhesus monkeys (monkey G: 8 years old, 17.5 Kg; monkey J: 7 years old, 16.5 Kg) were used in this study. We used the same animals and interface published in (*Yazdan-Shahmorad et al., 2016*). All procedures were performed under the approval of the University of California, San Francisco Institutional Animal Care and Use Committee and were compliant with the Guide for the Care and Use of Laboratory Animals.

### Optogenetic Interface

Here, we give a brief summary of our large-scale optogenetic interface (details in (*Yazdan-Shahmorad et al., 2016*). We combined three existing techniques to implement a practical, large-scale interface for both manipulation and recording from the surface of the brain in two rhesus macaque monkeys. We used an efficient technique for infusion of the optogenetics viral vector (AAV5.CamKIIa.C1V1(E122T/E162T).TS.eYFP.WPRE.hGH, $2.5 \times 10^{12}$ virus molecules/ml; Penn Vector Core, University of Pennsylvania, PA, USA, Addgene number: 35499) into primary somatosensory (S1) and motor (M1) cortices based on convection-enhanced delivery (CED). We infused 200 μL of virus in four sites (two in M1 and two in S1) in monkey G and 250 μL in five sites (two in M1 and three in S1) of monkey J. Infusion rate started at 1 μl/min and was increased to 5 μl/min. Following infusion we used an artificial dura to protect the brain while maintaining optical access. We verified expression with epifluorescent imaging. To record the evoked responses in both M1 and S1, we used either one or two 96 channel micro-electrocorticography (μ-ECoG) arrays that were designed to allow minimally-attenuated (*Ledochowitsch et al., 2015*) optical access. In both animals, we observed reliable light evoked neural responses from the large channel-expressing areas.

### Data acquisition

Optical stimulation was applied using a fiber optic (core/cladding diameter: 62.5/125 um, Fiber Systems, TX, USA) that was connected to a 488 nm laser (PhoxX 488–60, Omicron-Laserage, Germany) and was positioned above the array (*Figure 1*). We used a Tucker-Davis Technologies system (FL, USA) for μ-ECoG recording and to control the laser stimulus.

We chose the locations of stimulation based on the results of our epifluorescent imaging (see (*Yazdan-Shahmorad et al., 2016*)) and whether we got a physiological response to light stimulation. Based on detailed histological and electrophysiological analyses, (see (*Yazdan-Shahmorad et al., 2016*)) we estimate that there was a uniform distribution of expression around our infusion sites and at the locations that we chose to stimulate.

The monkeys were awake sitting in primate chairs for the duration of experiments. To ensure that the monkey remained awake, we provided random reward as well as audio and visual stimuli (displaying a cartoon). In addition, we visually monitored the animal throughout the experiment.

### Stimulation and recording protocol

Our stimulation and recording paradigms consisted of baseline recording, testing and conditioning blocks (*Figure 4A*). During baseline recording blocks we collected 5 min or 30 s of baseline activity. In test blocks we delivered 100 light pulses (five msec duration at 5 Hz) to each laser; pulses were alternated between laser sites in blocks of 10 pulses each in order to reduce the effects of habituation. For the conditioning blocks we stimulated through one or two lasers at a frequency of 5 or 7 Hz for 10 min. We repeated this protocol five times for each experiment. Our experiments consisted of single- and two-site stimulation with different laser configurations: i) one or two lasers in M1, ii) one or two lasers in S1 and iii) one laser in M1 and one laser in S1. For two-site stimulation, the delay between the two lasers was 10, 30, 70 or 100 ms. We also included control sessions in which we kept the structure of passive recording and active testing blocks, but did not stimulate during the conditioning blocks.

### Quantification and Statistical Analysis

#### Preprocessing

All processing and statistical analyses were performed using custom MATLAB (MA, USA) code. After signal acquisition, broadband surface potentials (sampled at 24 kHz) were visually inspected, and faulty recording sites were removed from further analysis. Next, stimulation-triggered responses

were visually inspected for photoelectric artifacts (characterized by their timing, and amplitude; see (*Yazdan-Shahmorad et al., 2016*; *Ledochowitsch et al., 2015*)), and removed from analysis. Next, two measures of connectivity were defined to quantify the strength of functional connectivity between M1 and S1.

## Stimulus-Evoked Response Ratio

During test blocks, we delivered 100 stimulus pulses at 5 Hz through each laser. To capture high-fidelity timing of the signal and avoid significant phase distortions we acausally band-pass filtered [0.1–500 Hz] the broadband signal. We first calculated the delay between the onset of the stimulus pulse and the response measured at each electrode. We calculated the delay from an average waveform across the distribution of evoked responses (bootstrapped 1000 times). This bootstrapped averaging was less susceptible to artifacts and noise that could dominate the signal on individual trials. The average evoked response lasted no longer than 30 ms with the trough of the response occurring in the first 5–15 ms. We defined the delay of the evoked response as the time of the trough within 20 ms of stimulation onset (*Figure 2A and E*). The distribution of delays across recording sites (*Figure 2B and F*) was bimodal with a delay greater than 1.5 ms between the modes. We manually set a threshold based on this delay distribution to distinguish between 'primary' and 'secondary' sites, i.e. those in the same area as the stimulation site and those in the other area. Sessions without a secondary response were excluded from further analysis (see *Table 1* for details).

Next, we calculated the amplitude of each evoked response. We downsampled the raw surface field potentials to 1 kHz (after applying a lowpass Chebychev filter for anti-aliasing) and applied an acausal band-pass filter to capture high gamma activity (60–200 Hz)), which is known to be representative of neural activity of local cortical columns (*Suzuki and Larkum, 2017*; *Yazdan-Shahmorad et al., 2013*). The peak (maximum) and trough (minimum) of the evoked waveform were identified within a 20 ms window after each laser pulse (*Figure 2AE*, in dashed inset boxes) and the peak-to-trough difference was computed. The SERR was then defined as the average (across 100 repeated laser pulses) of the ratio of two peak-to-trough amplitudes, the one for the secondary site over the one for the stimulation site. We considered other measures for calculating SERR, including commonly used measures such as the amplitude or slope of the broadband stimulus-evoked response (see *Table 2*). High gamma peak-to-trough amplitude yielded the most robust results.

Changes in SERR were calculated by taking the difference in SERR between the initial and final testing blocks for each secondary site. A paired, two-sided t-test was used to determine whether SERR changes were significant across secondary recording sites. Similarly, a paired, two-sided t-test was applied to determine significance across sessions separately for controls (in which no stimulation was applied during conditioning), stimulation in M1, and stimulation in S1. Unpaired, two-sided t-tests were used to directly compare the change in SERR between all stimulation sessions and all control sessions. P-values less than. 05 were considered significant.

## Coherence

First, broadband surface potentials were downsampled to 1 kHz (after applying a lowpass Chebychev filter for anti-aliasing). Then pairwise coherences were calculated in 10 s Hamming windows in 4 Hz frequency bands. The coherence between channels x and y is defined as:

**Table 1.** Summary of the sessions for both monkeys.
The data was collected in two to three week periods for each animal. Depending on the health of the animal and the quality of the neural recordings one to four experiments were performed per day.

|  | Monkey G | Monkey J |
| --- | --- | --- |
| Number of sessions | 37 | 33 |
| Number of sessions analyzed | 29 | 15 |
| Number of control sessions | 3 | 2 |
| Number of single-site and long latency sessions | 6 | 9 |
| Number of short-latency sessions | 20 | 4 |

DOI: https://doi.org/10.7554/eLife.31034.036

**Table 2.** Comparison between different measures for calculating SERR.

We decided a priori to calculate the SERR using the high gamma peak to trough as a response metric, since high gamma potentials are thought to represent the population activity of local cortical columns (*Yazdan-Shahmorad et al., 2013*; *Suzuki and Larkum, 2017*). Post-hoc, we performed further analysis to investigate whether the other measures (listed in the above table) show similar effects to the high-gamma peak to trough. As shown here, all of the measures show a significant increase following stimulation (second column) whereas they do not show a significant change after control sessions in which there is no stimulation applied (first column). However, other measures are more variable and do not show increases relative to the control sessions (third column). Furthermore, only high-gamma peak to trough and broadband amplitude are significantly correlated with coherence in the baseline condition (fourth column). Overall, this suggests that the amplitude of the high-gamma signal is the best of these metrics for estimating connectivity. All effect sizes in this table reflect the median change in connectivity across sessions for each measure; p-values reflect the output of signrank (columns 1 and 2) and ranksum (column 3) statistical tests.

| Evoked Response Measure | Change in Connectivity during Control Sessions | Change in Connectivity during Long Latency Stim Sessions | Change in Connectivity in Control Sessions Vs. Change in Connectivity in Stim Sessions | Correlation Between Coherence (Across Freqs) and Evoked Response Connectivity Measure |
|---|---|---|---|---|
| high-gamma peak to trough (SERR) | eff_size = 0.018 p=1.0 | eff_size = 0.118 p=1.6e-04 | eff_size = 0.100 p=6.6e-03 | avg_slope = 0.57 p=3.7e-06 |
| high-gamma energy | eff_size = 0.09 p=0.38 | eff_size = 0.86 p=3.3e-03 | eff_size = 0.77 p=0.184 | avg_slope = 0.037 p=0.144 |
| broadband energy | eff_size = 0.28 p=0.22 | eff_size = 0.39 p=5.4e-04 | eff_size = 0.12 p=0.386 | avg_slope = −0.02 p=0.652 |
| broadband amplitude | eff_size = 0.21 p=0.30 | eff_size = 0.23 p=4.6e-04 | eff_size = 0.01 p=0.184 | avg_slope = 0.183 p=0.002 |
| broadband slope | eff_size = 0.09 p=0.22 | eff_size = 0.23 p=4.6e-04 | eff_size = 0.13 p=0.184 | avg_slope = 0.028 p=0.318 |

DOI: https://doi.org/10.7554/eLife.31034.037

The following source data available for Table 2:

**Source code 1.** Statistics of SERR and other connectivity measures for each session in each experimental condition.
DOI: https://doi.org/10.7554/eLife.31034.038

**Source data 1.** SERR and other connectivity measures for each session in each experimental condition.
DOI: https://doi.org/10.7554/eLife.31034.039

$$C_{xy}(f) = \frac{|G_{xy}(f)|^2}{G_{xx}(f)G_{yy}(f)}$$

where $G_{xx}$ and $G_{yy}$ refer to power spectral density of channels x and y respectively, and $G_{xy}$ refers to their cross-spectral density. For simplicity, in future equations we refer to the coherence between channels x and y as C. Coherence in the theta band (4–8 Hz) was used in all analyses, unless otherwise specified.

Changes in inter-area coherence were calculated by taking the difference in coherence between the initial and final recording blocks. Significant changes after individual sessions were detected with a paired, two-sided t-test across secondary sites. Across sessions, changes in inter-area coherence in each frequency band were compared between stimulation sessions and control sessions with an unpaired, two-sided t-test, applying the Bonferroni correction for multiple comparisons.

## Connectivity Dynamics

Both the SERR and inter-area theta coherence were measured for each baseline recording and each test block during the experiment. To measure connectivity during the conditioning block, the block was sectioned into 100-pulse segments, and the SERR was calculated for each of these segments. To understand the connectivity trends, we pooled experiments together and calculated the average connectivity measures. To account for initial differences in connectivity across sessions and monkeys, we calculated the average changes in each connectivity measure with respect to its initial value and computed their Pearson correlation across sessions and blocks.

## Network Analysis

Since stimulation evokes a network-wide pattern of activity, the coherence during conditioning blocks ($C_c$) is different than the coherence calculated during baseline recording blocks ($C_r$). This difference, or 'stimulus-evoked coherence' captures the correlations introduced through stimulation. We propose that these correlations drive plasticity in a Hebbian manner, so that the change in baseline coherence between pre- and post-conditioning reflects the stimulus-evoked coherence. Therefore, we assessed how well the change in recording coherence is predicted, across blocks and electrode pairs, by the stimulus-evoked coherence using linear regression,

$$\left(C_c - C_{r,pre}\right) = \hat{B} * \left(C_{r,post} - C_{r,pre}\right),$$

where $B$ are the fit regression parameters. However, this simple regression analysis is biased, since the same $C_r$ is used on the left and right side of this equation. To avoid the spurious correlations that would therefore arise, we split the recording blocks into 5s non-overlapping windows and calculated the coherence separately for the odd (α) and even (β) windows. We then averaged the coherence for each set of windows ($C_{r,\alpha}$ and $C_{r,\beta}$) and used these as independent measures of baseline coherence for the regression,

$$\left(C_c - C_{r,pre,\alpha}\right) = \hat{B} * \left(C_{r,post,\alpha} - C_{r,pre,\beta}\right).$$

## Histology analysis

Monkeys were deeply sedated (per above Surgical procedures) and perfused transcardially with heparinized phosphate buffered saline (PBS) followed by cold 4% paraformaldehyde in phosphate buffer. The brain was extracted and post-fixed in the same fixative for 24 hr at 4°C and then dissected into twelve 6 mm-thick coronal blocks using a custom matrix. After 7–10 days incubation in 30% sucrose, blocks were frozen and cut on a cryostat (Microm, Germany) into 50 μm thick sections. Representative sections were selected from each block and processed for EYFP immunocytochemistry using a free-floating technique. Sections were initially washed in PBS, incubated in 3% hydrogen peroxide in PBS for 10 min to quench endogenous peroxidase activity, then rinsed in two changes of 50% ethanol followed by three changes of PBS for 5 min each. Next we incubated the sections in 5% normal donkey serum in PBS for 1 hr to block non-specific binding. Primary rabbit polyclonal anti-GFP antibody (Abcam, RRID: AB_303395) was diluted 1:15,000 in PBS containing 0.01% Triton X-100 and was applied to the tissue for 48 hr at 4°C. Sections were then rinsed in PBS, incubated in biotinylated donkey anti-rabbit antibody, (1:2,000, Jackson Immunoresearch) for 12 hr at 4°C, rinsed and incubated in ExtrAvidin (Sigma-Aldrich, 1:5,000) for 5–6 hr in room temperature. Peroxidase was detected using a diaminobenzidine (DAB) chromogenic reagent (Sigma-Aldrich). Sections were rinsed in PBS, mounted on gelatin-coated slides, air dried, dehydrated in graded alcohols, cleared in xylene and coverslipped with D.P.X. mounting media (Sigma-Aldrich). Additional adjacent sections were stained with cresyl violet (Nissl) using standard techniques, to reveal cortical cytoarchitecture.

Double immunofluorescence was performed using the similar to above approach although using combinations of primary antibodies from different host species for GFP and for the interneuron markers. Primary antibodies were: rabbit polyclonal anti-GFP antibody (1:10,000, Abcam, GFP ab290); goat polyclonal anti-GFP antibody (1:1,000, Abcam, ab5450). These were paired with one of the following primary antibodies: mouse monoclonal antibody for parvalbumin, (1:1,000, Sigma-Aldrich P3088); mouse monoclonal antibody for calbindin (1:800 Sigma-Aldrich CB-955); rat monoclonal antibody for somatostatin (1:200, Millipore MAP354); rabbit polyclonal anti-GAD65/67 (1:500, Millipore AB1511); a cocktail of mouse monoclonal antibody for GAD67 (1:500, Millipore MAB5406) and mouse monoclonal antibody for GAD65 (1:1,000, Sigma-Aldrich G1166). Sections were blocked using normal donkey serum and processed for 4–6 hr in a mix of matching secondary antibodies; all of which were raised in donkey (Thermo Fisher Scientific, 1:300): anti-rabbit Alexa Fluor 488, anti-goat Alexa Fluor 488 (both for GFP). Interneuron markers were visualized using anti-mouse Alexa Fluor 594, anti-rat Alexa Fluor 594; anti-rabbit Alexa Fluor 594. Sections were mounted using Vectashield (Vector Labs) and imaged using Zeiss LSM510 Meta confocal microscope (Zeiss, Germany) and 90i imaging system equipped with a CCD camera (Nikon, Japan).

## Acknowledgements

We thank Lindsey Presson, Marc Meneses, and Julien Reichenmann for their help with the animals, Camilo Diaz-Botia for device fabrication, Kate Derossier for her help with the experiments, Eberhard Fetz for helpful comments on the manuscript and Ali Shojaee for advice on statistical analysis.

## Additional information

### Competing interests

Philip N Sabes: has financial interest in Neuralink Corp., a company that is developing clinical therapies using brain stimulation. The other authors declare that no competing interests exist.

### Funding

| Funder | Grant reference number | Author |
| --- | --- | --- |
| Defense Advanced Research Projects Agency | W911NF-14-2-0043 | Azadeh Yazdan-Shahmorad<br>Daniel B Silversmith<br>Viktor Kharazia<br>Philip N Sabes |
| American Heart Association | Post-doctoral fellowship | Azadeh Yazdan-Shahmorad |
| National Science Foundation | Graduate student fellowship | Daniel B Silversmith |

This research was partially funded by the Defense Advanced Research Projects Agency (DARPA) under Cooperative Agreement Number W911NF-14-2-0043, issued by the Army Research Office contracting office in support of DARPA'S SUBNETS program. The views, opinions, and/or findings expressed are those of the author(s) and should not be interpreted as representing the official views or policies of the Department of Defense or the U.S. Government. The funders had no role in study design, data collection and interpretation, or the decision to submit the work for publication.

### Author contributions

Azadeh Yazdan-Shahmorad, Daniel B Silversmith, Conceptualization, Data curation, Software, Formal analysis, Funding acquisition, Validation, Visualization, Methodology, Writing—original draft, Writing—review and editing; Viktor Kharazia, Resources, Data curation, Formal analysis, Validation, Visualization, Methodology, Writing—review and editing; Philip N Sabes, Conceptualization, Resources, Supervision, Funding acquisition, Validation, Methodology, Project administration, Writing—review and editing

### Author ORCIDs

Azadeh Yazdan-Shahmorad (iD) http://orcid.org/0000-0001-5212-509X
Daniel B Silversmith (iD) http://orcid.org/0000-0003-1771-1856
Philip N Sabes (iD) https://orcid.org/0000-0001-8397-6225

### Ethics

Animal experimentation: All procedures were performed under the approval of the University of California, San Francisco Institutional Animal Care and Use Committee (AN108552-03) and were compliant with the Guide for the Care and Use of Laboratory Animals.

### Decision letter and Author response

Decision letter https://doi.org/10.7554/eLife.31034.045
Author response https://doi.org/10.7554/eLife.31034.046

## Additional files

### Supplementary files
• Supplementary file 1. Spreadsheet of statistical information
DOI: https://doi.org/10.7554/eLife.31034.040
• Transparent reporting form
DOI: https://doi.org/10.7554/eLife.31034.041

### Data availability
We have provided the numerical data (.mat format) for all of the graphs in all of the figures except where images or raw data were presented. For each figure we are providing ReadMe files that include descriptions of the parameters used as well as the Matlab code for generating the figures. In addition, we have made the full dataset available via UCSF data share program: https://datashare.berkeley.edu/stash/dataset/doi:10.7272/Q61834NF.

The following dataset was generated:

| Author(s) | Year | Dataset title | Dataset URL | Database, license, and accessibility information |
|---|---|---|---|---|
| Yazdan-Shahmorad A, Silversmith DB, Kharazia V, Sabes PN | 2018 | Targeted Cortical Reorganization using Optogenetics in Non-human primates: Electrocorticography in Sensorimotor Cortex during Optogenetic Stimulation | https://dash.berkeley.edu/stash/dataset/doi:10.7272/Q61834NF | Available at Dash under a Creative Commons Attribution 4.0 International (CC BY 4.0) license |

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
