## [Decision Letter]

Thank you for submitting your article "Targeted cortical reorganization using optogenetics in non-human primates" for consideration by *eLife*. Your article has been favorably evaluated by Sabine Kastner (Senior Editor) and three reviewers, one of whom, Charles Schroeder (Reviewer #1), is a member of our Board of Reviewing Editors. The following individuals involved in review of your submission have agreed to reveal their identity: Luke Sjulson (Reviewer #2); Michael Michaelides (Reviewer #3).

The reviewers have discussed the reviews with one another and the Reviewing Editor has drafted this decision to help you prepare a revised submission.

Summary:

This paper reports on investigation of stimulation-induced changes in intracortical functional connectivity in 2 monkeys. S1 and M1 were studied with dense arrays of micro-ECoG electrodes positioned at the Pial surfaces of the areas. Stimulation-related changes in both "high-gamma" band activity (~related to neuronal firing) and changes in resting theta band coherence at rest were examined. Stimulation was provided by optogenetic methods that were designed to be selective to excitatory (pyramidal) neurons. The experiments are a significant technical achievement, and the idea of evoking and measuring plasticity between two brain regions is interesting. The main effect of periods of conditioning stimulation was a general increase in connectivity by all measures, though it had heterogeneous characteristics whose underpinnings were unclear. The authors conclude that their findings generalize the concept of Hebbian plasticity to the network scale and that this has clinical implications. Despite the paper's strengths, numerous problems prevent the main scientific result of optogenetic Hebbian plasticity from being established clearly. Additionally, parts of the manuscript are sufficiently confusing that it was difficult to understand what the experimental conditions were in order to evaluate them.

Essential revisions:

1) "Traditional" somatosensory plasticity work in monkeys (Merzenich, Kaas, Garraghty, etc.) relates to re-organization after input change due to injury or some other perturbation. How does the present study interface with that extensive body of findings? Notably the somatosensory reorganization work focuses on the somatotopic representation in S1, but that is not dealt with at all here (see next point) noted, the somatosensory and motor system are already interconnected, but within area there is a somatotopic organization, and across areas, connectivity tends to be between ~ corresponding somatotopic representations. It would add greatly to the clarity of this paper if this point were addressed.

2) No mention of the monkeys' state of anesthesia is made here. Presumably they were anesthetized. Regardless, that issue point be clearly discussed, as one would reasonably expect that the monkeys' state would have a large impact on the plasticity of the system.

3) SERR is a critical measure for this paper, and it needs to be described and illustrated in much more detail – see below Figure 2 also. In particular, why is the peak to trough the measure of interest, rather than, e.g., power in the 60-200 Hz band over a defined post-stimulation epoch. Additionally, if the lag of the response peak in the secondary response (Figure 2A-B) is important in that case, why not here as well? How exactly were the peaks and troughs scored and how is peak to trough amplitude measured?

4) The manuscript tacitly promotes a description of the experiment that may not be correct, in which excitatory cells in S1 and M1 are stimulated independently. In reality, there are substantial projections from S1 to M1 and vice versa, and illumination of either structure would be expected to evoke action potentials in both the illuminated somata and the axon terminals of neurons from the other structure. Xiong et al. (https://www.ncbi.nlm.nih.gov/pubmed/25731173) and others have measured this effect in vivo using techniques similar to those used in this manuscript. One feature of C1V1 that may work in the authors' favor is that it traffics poorly to long-range projections (https://www.ncbi.nlm.nih.gov/pubmed/26436451), but nevertheless, the authors must address this point. One suggestion would be to break down the temporal components of the LFP responses as in Xiong et al.

5) In Figure 4, the authors show comparisons with a "No Stim" condition. However, the no stim condition is not plotted in 4G or any of the subsequent figures in the paper. Additionally, the various statistical tests for increased connectivity are not compared against the no stim condition, as they should be.

6) I think the main claim being made by this paper is that correlated stimulation causes strengthening of functional connectivity, and uncorrelated stimulation weakens it. However, the paper is sufficiently unclear that it is difficult to determine whether that is in fact the claim the authors intend to make. Figure 1B shows only one protocol, but the Materials and methods section lists multiple protocols in incomprehensible fashion. This manuscript needs a transparent explanation of what the experimental conditions were, and Figures 6-7 need to contain direct statistical comparisons of these conditions against each other (including the "no stim" condition). Ideally all of the experimental conditions should be illustrated as in Figure 1B.

7) In their 2016 publication, the authors report the use of an AAV5 C1V1 virus. Here they report a pseudotyped AAV2/5. Are these the same virus? If not why did the authors switch. Neither show up on the UPENN vector core website catalog. Were they custom made? This is important information that should be shared so that other research groups are able to perform similar experiments if they so wish to. Also in the 2016 publication, the authors reported that the artificial dura led to anatomical changes in the targeted area. The authors used the same approach in these experiments but any such effects or measures were not reported. Did they observe such changes here?

8) The authors show convincing and high quality immunohistochemical data however they did not report whether they attempted to quantify the number of cells that expressed the opsin in each monkey, especially at the site of stimulation and whether this was related to their network connectivity measures. It would have been useful (but not necessary at this point if they do not have this data) if they had chosen to stimulate in different regions and then examined their downstream measures as a function of cell transduction. In relation to the above point, how did the authors decide on the specific areas within each region to stimulate? Was this based on where the injections were made?

---

## [Author Response]

Essential revisions:1) "Traditional" somatosensory plasticity work in monkeys (Merzenich, Kaas, Garraghty, etc.) relates to re-organization after input change due to injury or some other perturbation. How does the present study interface with that extensive body of findings? Notably the somatosensory reorganization work focuses on the somatotopic representation in S1, but that is not dealt with at all here (see next point) noted, the somatosensory and motor system are already interconnected, but within area there is a somatotopic organization, and across areas, connectivity tends to be between ~ corresponding somatotopic representations. It would add greatly to the clarity of this paper if this point were addressed.

Unfortunately due to time constraints in running these experiments, we did not obtain somatotopic maps for the cortical regions under study. These animals have subsequently been sacrificed to obtain histological data.

As the reviewers note, there is a great body of research that focuses on sensory cortical plasticity; we see this paper as complementary. While previous work has largely focused on how changes in the statistics of sensory input and/or motor output affect the cortical representation in sensory cortex, we have directly manipulated cortical circuits, irrespective of their somatotopic map. And where previous work studied plasticity on the time scale of days to months, we are studying changes across minutes. Despite the substantial differences between, however, both appear to support a key role for Hebbian learning in sensorimotor cortical plasticity. Still, our ability to link the present work to previous work is clearly limited by the lack of cortical mapping and the inability to study changes on the intermediate times scales. These are clearly important questions for future work. We have added these ideas to the Discussion.

2) No mention of the monkeys' state of anesthesia is made here. Presumably they were anesthetized. Regardless, that issue point be clearly discussed, as one would reasonably expect that the monkeys' state would have a large impact on the plasticity of the system.

We thank the reviewers for pointing out this omission. In fact, the monkeys were awake, sitting in primate chairs, and watching enrichment videos (cartoons) during these experiments. To ensure that the monkey stayed awake for the duration of the experiment we provided random reward as well as audio and visual stimuli. In addition we monitored the monkeys via a camera. We have added this information to Materials and methods.

3) SERR is a critical measure for this paper, and it needs to be described and illustrated in much more detail – see below Figure 2 also. In particular, why is the peak to trough the measure of interest, rather than, e.g., power in the 60-200 Hz band over a defined post-stimulation epoch. Additionally, if the lag of the response peak in the secondary response (Figure 2A-B) is important in that case, why not here as well? How exactly were the peaks and troughs scored and how is peak to trough amplitude measured?

A priori, we chose to measure SERR with the high gamma amplitude (60-200 Hz, peak-to-trough) as a response metric, since high gamma potentials are thought to represent the population activity of local cortical columns. Per the reviewers’ suggestion, we re-evaluated our results based on several different measures extracted from the evoked responses. These measures include energy in the 60-200 Hz (high-gamma) band, energy of the broadband signal, amplitude of the broadband signal and the slope of the broadband signal. All of these measures showed similar changes after stimulation, though only the original measure exhibited a significant difference between stimulation and control sessions. More generally, the high gamma amplitude yielded the most robust results and best correlated with coherence. These data and a discussion of them were added to Table 2. We have also added more detailed explanation of how the peaks and troughs are measured. Furthermore, we modified Figure 2 and the text to explain the SERR better.

4) The manuscript tacitly promotes a description of the experiment that may not be correct, in which excitatory cells in S1 and M1 are stimulated independently. In reality, there are substantial projections from S1 to M1 and vice versa, and illumination of either structure would be expected to evoke action potentials in both the illuminated somata and the axon terminals of neurons from the other structure. Xiong et al. (https://www.ncbi.nlm.nih.gov/pubmed/25731173) and others have measured this effect in vivo using techniques similar to those used in this manuscript. One feature of C1V1 that may work in the authors' favor is that it traffics poorly to long-range projections (https://www.ncbi.nlm.nih.gov/pubmed/26436451), but nevertheless, the authors must address this point. One suggestion would be to break down the temporal components of the LFP responses as in Xiong et al.

The Xiong et al. paper was able to parse the timing of the evoked response at a given site in order to evaluate synaptic efficacy because they were stimulating axon terminals and recording cells at the stimulation site. This technique cannot be applied to our paradigm, where we are stimulating at the (nominal) upstream site and recording at the (nominal) downstream site, since the temporal effects will be washed out by a variety of factors, including transduction and spike generation at the stimulation site and axonal conduction to the recording site.

In fact, we are likely evoking activity from both terminals and cell bodies at the stimulation site, though as the referee notes, C1V1 does not express well down the axon. Therefore, our secondary responses arise from a combination of synaptic, antridromic, and indirect (network) effects. That said, we do not expect the antidromic effects to contribute to stimulation-induced plastic changes in the response. And the effect of larger-scale network contributions are discussed in the last section of the Results. We have added a section on this topic to the Discussion.

5) In Figure 4, the authors show comparisons with a "No Stim" condition. However, the no stim condition is not plotted in 4G or any of the subsequent figures in the paper. Additionally, the various statistical tests for increased connectivity are not compared against the no stim condition, as they should be.

We have added figures to the supplementary material that compares no-stim to the results presented for Figures 4 and 5. Also we have added statistical tests and results to the text to compare the changes following stimulation with the no-stim condition. As expected, the control sessions do not show the plasticity effects that are the main result of the paper.

6) I think the main claim being made by this paper is that correlated stimulation causes strengthening of functional connectivity, and uncorrelated stimulation weakens it. However, the paper is sufficiently unclear that it is difficult to determine whether that is in fact the claim the authors intend to make. Figure 1B shows only one protocol, but the Materials and methods section lists multiple protocols in incomprehensible fashion. This manuscript needs a transparent explanation of what the experimental conditions were, and Figures 6-7 need to contain direct statistical comparisons of these conditions against each other (including the "no stim" condition). Ideally all of the experimental conditions should be illustrated as in Figure 1B.

The main claims of the paper are: 1) Stimulating at one site drives correlated activity that strengthens functional connectivity between brain areas. 2) Given stimulus-evoked correlated activity, a Hebbian model predicts changes in functional connectivity. 3) This model holds even when complex patterns of stimulation drive a combination of strengthening and weakening of connectivity. We have tried to make these claims clearer by changing our section and figure titles and by making clarifying edits throughout the paper.

In addition, we have modified Figure 1 and added schematics of the stimulation protocols to Figures 4 and 7 to better explain the stimulation protocols. In addition we have done direct statistical analysis between “no stim” and stim data for Figures 6 and 7. The results of these comparisons are added to the text and Supplementary file 1.

7) In their 2016 publication, the authors report the use of an AAV5 C1V1 virus. Here they report a pseudotyped AAV2/5. Are these the same virus? If not why did the authors switch. Neither show up on the UPENN vector core website catalog. Were they custom made? This is important information that should be shared so that other research groups are able to perform similar experiments if they so wish to. Also in the 2016 publication, the authors reported that the artificial dura led to anatomical changes in the targeted area. The authors used the same approach in these experiments but any such effects or measures were not reported. Did they observe such changes here?

We thank the referee for pointing out this omission. In fact, the data presented here are from the same animals as the 2016 Neuron paper, using the same interfaces and the same cortical areas. The virus (AAV5.CamKIIa.C1V1(E122T/E162T).TS.eYFP.WPRE.hGH) was custom made by UPENN vector core and the Addgene number is 35499. We have added this information to the text.

As the reviewer notes, we did see some cortical thinning in monkey G post-mortem (as reported in our 2016 Neuron paper). In contrast, the histology for monkey J, presented here does not show change in the cortical thickness (see Figure 1—figure supplement 1A-F). This difference has been added to Materials and methods and discussed in the supplement.

8) The authors show convincing and high quality immunohistochemical data however they did not report whether they attempted to quantify the number of cells that expressed the opsin in each monkey, especially at the site of stimulation and whether this was related to their network connectivity measures. It would have been useful (but not necessary at this point if they do not have this data) if they had chosen to stimulate in different regions and then examined their downstream measures as a function of cell transduction. In relation to the above point, how did the authors decide on the specific areas within each region to stimulate? Was this based on where the injections were made?

Please see our 2016 Neuron paper (Yazdan-Shahmorad, et al., 2016) for more detailed report on the optogenetic expression level for Monkey G. We chose the locations of stimulation to be within regions of high expression, as assessed with epifluorescent imaging (see Yazdan-Shahmorad, et al., 2016). Based on our histological and electrophysiological analyses (see Yazdan-Shahmorad, et al., 2016), we concluded that there was a fairly uniform distribution of expression around our infusion sites, including the locations that we chose to stimulate for this paper. We have added this information to the paper.